# Implicit Behavioral Cloning

**Pete Florence, Corey Lynch, Andy Zeng, Oscar Ramirez, Ayzaan Wahid,
Laura Downs, Adrian Wong, Johnny Lee, Igor Mordatch, Jonathan Tompson**

Robotics at Google

**Abstract:** We find that across a wide range of robot policy learning scenarios, treating supervised policy learning with an *implicit model* generally performs better, on average, than commonly used explicit models. We present extensive experiments on this finding, and we provide both intuitive insight and theoretical arguments distinguishing the properties of implicit models compared to their explicit counterparts, particularly with respect to approximating complex, potentially discontinuous and multi-valued (set-valued) functions. On robotic policy learning tasks we show that implicit behavioral cloning policies with energy-based models (EBM) often outperform common explicit (Mean Square Error, or Mixture Density) behavioral cloning policies, including on tasks with high-dimensional action spaces and visual image inputs. We find these policies provide competitive results or outperform state-of-the-art offline reinforcement learning methods on the challenging human-expert tasks from the D4RL benchmark suite, despite using no reward information. In the real world, robots with implicit policies can learn complex and remarkably subtle behaviors on contact-rich tasks from human demonstrations, including tasks with high combinatorial complexity and tasks requiring 1mm precision.

**Keywords:** Implicit Models, Energy-Based Models, Imitation Learning

## 1 Introduction

Behavioral cloning (BC) [1] remains one of the simplest machine learning methods to acquire robotic skills in the real world. BC casts the imitation of expert demonstrations as a supervised learning problem, and despite valid concerns (both empirical and theoretical) about its shortcomings (e.g., compounding errors [2, 3]), in practice it enables some of the most compelling results of real robots generalizing complex behaviors to new unstructured scenarios [4, 5, 6]. Although considerable research has been devoted to developing new imitation learning methods [7, 8, 9] to address BC's known limitations, here we investigate a fundamental design decision that has largely been overlooked: the form of the policy itself. Like many other supervised learning methods, BC policies are often represented by explicit continuous feed-forward models (e.g., deep networks) of the form $\hat{\mathbf{a}} = F_\theta(\mathbf{o})$ that map directly from input observations $\mathbf{o}$ to output actions $\mathbf{a} \in \mathcal{A}$. But what if $F_\theta$ is the wrong choice?

In this work, we propose to reformulate BC using *implicit models* – specifically, the composition of argmin with a continuous energy function $E_\theta$ (see Sec. 2 for definition) to represent the policy $\pi_\theta$:

$$\hat{\mathbf{a}} = \underset{\mathbf{a} \in \mathcal{A}}{\operatorname{argmin}} \; E_\theta(\mathbf{o}, \mathbf{a}) \qquad \text{instead of} \qquad \hat{\mathbf{a}} = F_\theta(\mathbf{o}) \; .$$

This formulates imitation as a conditional energy-based modeling (EBM) problem [10] (Fig. 1), and at inference time (given $\mathbf{o}$) performs implicit regression by optimizing for the optimal action $\hat{\mathbf{a}}$ via sampling or gradient descent [11, 12]. While implicit models have been used as partial components (e.g., value functions) for various reinforcement learning (RL) methods [13, 14, 15, 16], our work presents a distinct yet simple method: do BC with implicit models. Further, this enables a unique case study that investigates the choice between implicit vs. explicit policies that may inform other policy learning settings beyond BC.

Our experiments show that this simple change can lead to remarkable improvements in performance across a wide range of contact-rich tasks: from bi-manually scooping piles of small objects into bowls with spatulas, to precisely pushing blocks into fixtures with tight 1mm tolerances, to sorting mixed collections of blocks by their colors. Results show that implicit models for BC exhibit the capacity to learn long-horizon, closed-loop visuomotor tasks better than their explicit counterparts – and surprisingly, give rise to a new class of BC baselines that are competitive with state-of-the-art offline RL algorithms on standard simulated

5th Conference on Robot Learning (CoRL 2021), London, UK.

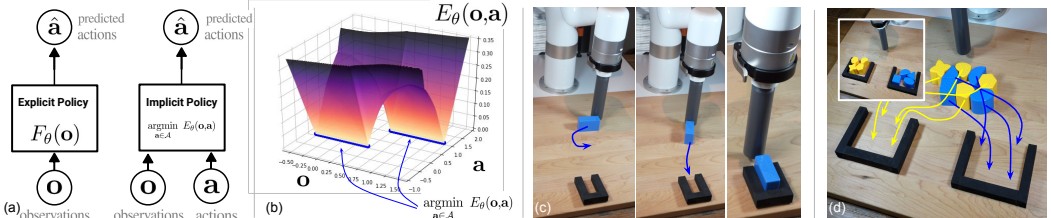

Figure 1. (a) In contrast to explicit policies, implicit policies leverage parameterized energy functions that take both observations (e.g. images) and actions as inputs, and optimize for actions that minimize the energy landscape (b). For learning complex, closed-loop, multimodal visuomotor tasks such as precise block insertion (c) and sorting (d) from human demonstrations, implicit policies perform substantially better than explicit ones.

benchmarks [17]. To shed light on these results, we provide observations on the intuitive properties of implicit models, and present theoretical justification that we believe are highly relevant to part of their success: their ability to represent not only multi-modal distributions, but also discontinuous functions.

**Paper Organization.** After a brief background (Sec. 2), to build intuition on the nature of implicit models, we present their empirical properties (Sec. 3). We then present our main results with policy learning (Sec. 4), both in simulated tasks and in the real world. Inspired by these results, we provide theoretical insight (Sec. 5), followed by related work (Sec. 6) and conclusions (Sec. 7).

## 2 Background: Implicit Model Training and Inference

We define an *implicit model* as any composition $(\arg\min_{\mathbf{y}} \circ E_\theta(\mathbf{x},\mathbf{y}))$, in which inference is performed using some general-purpose function approximator $E : \mathbb{R}^{m+n} \to \mathbb{R}^1$ to solve the optimization problem $\hat{\mathbf{y}} = \arg\min_{\mathbf{y}} E_\theta(\mathbf{x},\mathbf{y})$. We use techniques from the energy-based model (EBM) literature to train such a model. Given a dataset of samples $\{\mathbf{x}_i,\mathbf{y}_i\}$, and regression bounds $\mathbf{y}_{\min},\mathbf{y}_{\max} \in \mathbb{R}^m$, training consists of generating a set of negative counter-examples $\{\tilde{\mathbf{y}}_i^j\}_{j=1}^{N_{\text{neg.}}}$ for each sample $\mathbf{x}_i$ in a batch, and employing an InfoNCE-style [18] loss function. This loss equates to the negative log likelihood of $p_\theta(\mathbf{y}|\mathbf{x}) = \frac{\exp(-E_\theta(\mathbf{x},\mathbf{y}))}{Z(\mathbf{x},\theta)}$, and the counter-examples are used to estimate $Z(\mathbf{x}_i,\theta)$:

$$\mathcal{L}_{\text{InfoNCE}} = \sum_{i=1}^{N} -\log\big(\tilde{p}_\theta(\mathbf{y}_i | \mathbf{x}, \{\tilde{\mathbf{y}}_i^j\}_{j=1}^{N_{\text{neg.}}})\big), \quad \tilde{p}_\theta(\mathbf{y}_i | \mathbf{x}, \{\tilde{\mathbf{y}}_i^j\}_{j=1}^{N_{\text{neg.}}}) = \frac{e^{-E_\theta(\mathbf{x}_i,\mathbf{y}_i)}}{e^{-E_\theta(\mathbf{x}_i,\mathbf{y}_i)} + \sum_{j=1}^{N_{\text{neg}}} e^{-E_\theta(\mathbf{x}_i,\tilde{\mathbf{y}}_i^j)}}$$

With a trained energy model $E_\theta(\mathbf{x},\mathbf{y})$, implicit inference can be performed with stochastic optimization to solve $\hat{\mathbf{y}} = \arg\min_{\mathbf{y}} E_\theta(\mathbf{x},\mathbf{y})$. To demonstrate a breadth of approaches, we present results with three different EBM training and inference methods discussed below, however a comprehensive comparison of all EBM variants is outside the scope of this paper; see [19] for a comprehensive reference. We use either a) a derivative-free (sampling-based) optimization procedure, b) an auto-regressive variant of the derivative-free optimizer which performs coordinate descent, or c) gradient-based Langevin sampling [11, 12] with gradient penalty [20] loss during training – see the Appendix for descriptions and comparisons of these choices.

## 3 Intriguing Properties of Implicit vs. Explicit Models

Consider an explicit model $\mathbf{y} = f_\theta(\mathbf{x})$, and an implicit model $\arg\min_{\mathbf{y}} E_\theta(\mathbf{x},\mathbf{y})$ where both $f_\theta(\cdot)$ and $E_\theta(\cdot)$ are represented by almost-identical network architectures. Comparing these models, we examine: (i) how do they perform near discontinuities?, (ii) how do they fit multi-valued functions?, and (iii) how do they extrapolate? For both $f_\theta$ and $E_\theta$ we use almost-identical ReLU-activation fully-connected Multi-Layer Perceptrons (MLPs), with the only difference being the additional input of $\mathbf{y}$ in the latter. Explicit "MSE" models are trained with Mean Square Error (MSE), explicit "MDN" models are Mixture Density Networks (MDN) [21], and implicit "EBM" models are trained with $\mathcal{L}_{\text{InfoNCE}}$ and optimized with derivative-free optimization. Figs. 2, 3 show models trained on a number of $\mathbb{R}^1 \to \mathbb{R}^1$ functions (Fig. 2) and multi-valued functions (Fig. 3). For each of these we examine regions of discontinuities, multi-modalities, and/or extrapolation.

**Discontinuities.** Implicit models are able to approximate discontinuities sharply without introducing intermediate artifacts (Fig. 2a), whereas explicit models (Fig. 2d), because they fit a continuous function to the data, take every intermediate value between training samples. As the frequency of discontinuities increases, the implicit model predictions remain sharp at discontinuities, while also respecting local continuities, and with piece-wise linear extrapolations up to some decision boundary between training

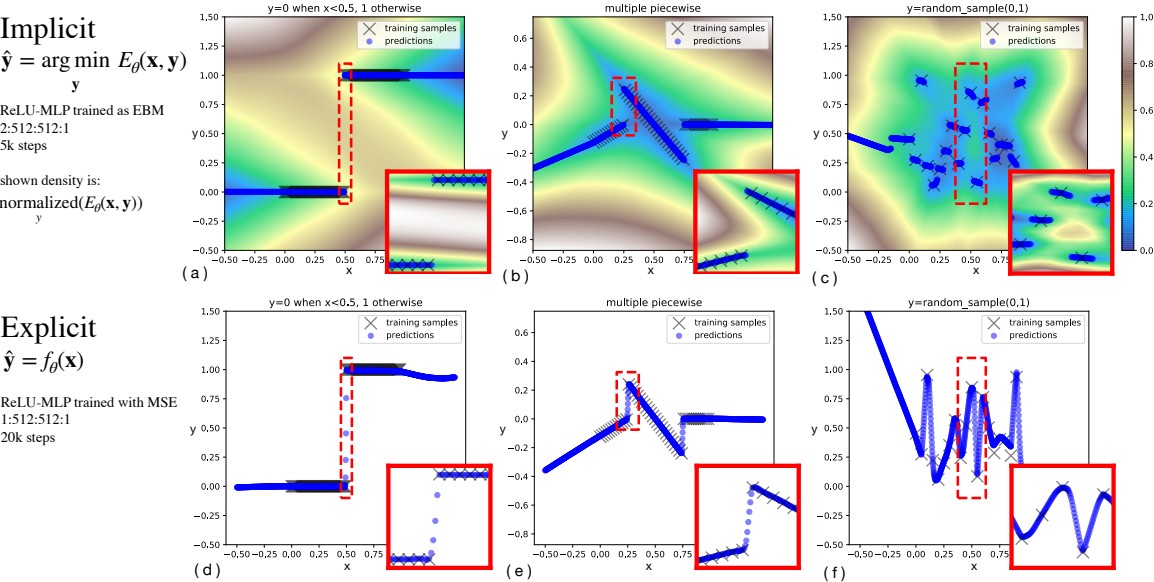

Figure 2. Comparison between implicit vs explicit learning of 1D functions, $\mathbb{R}^1 \to \mathbb{R}^1$, showing extrapolation (outside of $x = [0,1]$) behavior beyond training samples and detailed views (red insets) of interpolation behavior at discontinuities. (a,d) Single discontinuity between constant values; (b,e) piecewise continuous sections with differing $\frac{dy}{dx}$, (c,f) random Gaussian noise, for unregularized models.

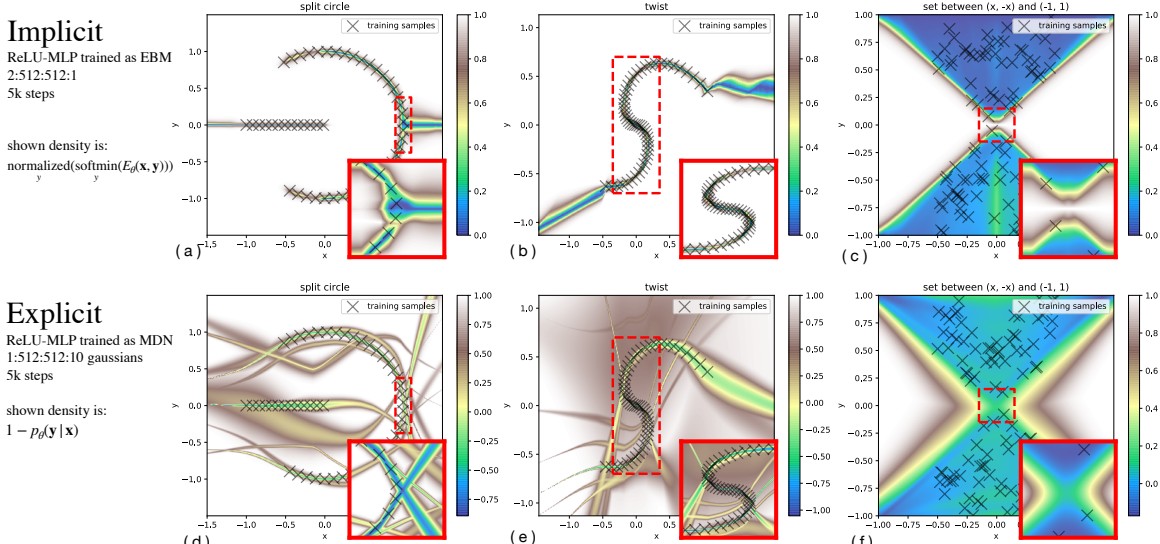

Figure 3. Representations of multi-valued functions showing extrapolations beyond the training samples (outside of shown 'X' training samples) and detail views of notable regions. (a,d) Split circle with discontinuities and mode count changes; (b,e) locally continuous curve exhibiting hysteretic behavior, (c,f) set function of disjoint uniformly valid ranges.

examples (Fig. 2a-c). The explicit model interpolates across each discontinuity (Fig. 2d-f). Once the training data is uncorrelated (i.e. random noise) and without regularization (Fig. 2c, Fig. 2f), implicit models exhibit a nearest-neighbors-like behavior, though with non-zero $\frac{\partial y}{\partial x}$ segments around each sample.

**Extrapolation.** For extrapolation outside the convex hull of the training data (Fig. 2a-f), even with discontinuous or multi-valued functions, implicit models often perform piecewise linear extrapolation of the piecewise linear portion of the model nearest to the edge of the training data domain. Recent work [22] has shown that explicit models tend to perform linear extrapolation, but the analysis assumes the ground truth function is continuous.

**Multi-valued functions.** Instead of using argmin to identify a single optimal value, argmin may return a set of values, which may either be interpreted probabilistically as sampling likely values from the distribution, or in optimization as the *set* of minimizers (argmin is set-valued). Fig. 3 compares a ReLU-MLP trained as a Mixture Density Network (MDN) vs an EBM across three example multi-valued functions.

**Visual Generalization** Of particular relevance to learning visuomotor policies, we also find striking differences in extrapolation ability with converting high-dimensional image inputs into continuous outputs. Fig. 4 shows how on a simple visual coordinate regression task, which is a notoriously hard problem for convolutional networks [23], an MSE-trained Conv-MLP model [24] with CoordConv [23] struggles to extrapolate outside the convex hull of the training data. This is consistent with findings in [5, 25]. A Conv-MLP trained via late fusion (Fig. 4b) as an EBM, on the other hand, extrapolates well with only a few training data samples, achieving 2 to 3 orders of magnitude lower test-set error in the low-data regime (Fig. 4d). This is additional evidence that distinguishes implicit models from explicit models in a distinct way from multi-modality, which is absent in this experiment.

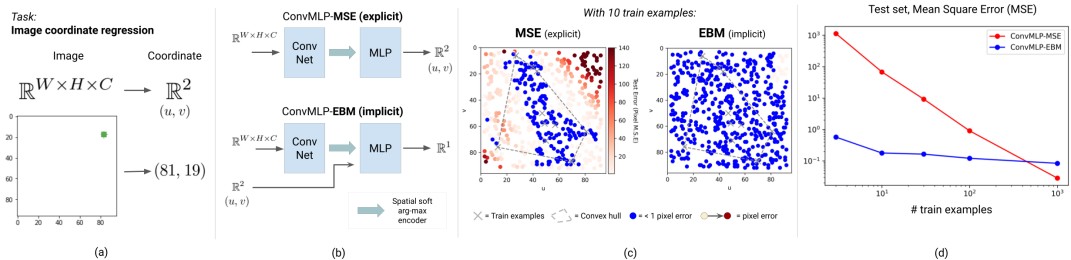

*Figure 4.* Comparison of implicit and explicit ConvMLP models on a simple coordinate regression task [23], $\mathbb{R}^{W \times H \times C} \to \mathbb{R}^2$ (a). The architectures shown in (b) are trained on images (example in a) to regress the $(u, v)$ coordinate of a green few-pixel dot. The *spatial generalization* plot (c) shows the convex hull (gray dotted line) of the training data and shows that with only 10 training examples, the MSE-trained models struggles to generalize, particularly outside of the convex hull (c, left). ConvMLP-EBM, instead (c, right) performs well with little data, with 2 to 3 orders of magnitude lower test-set MSE loss (d) in the low-data regime. With sufficient data as a dense sampling, both models perform well, with sub-pixel errors (d).

# 4    Policy Learning Results

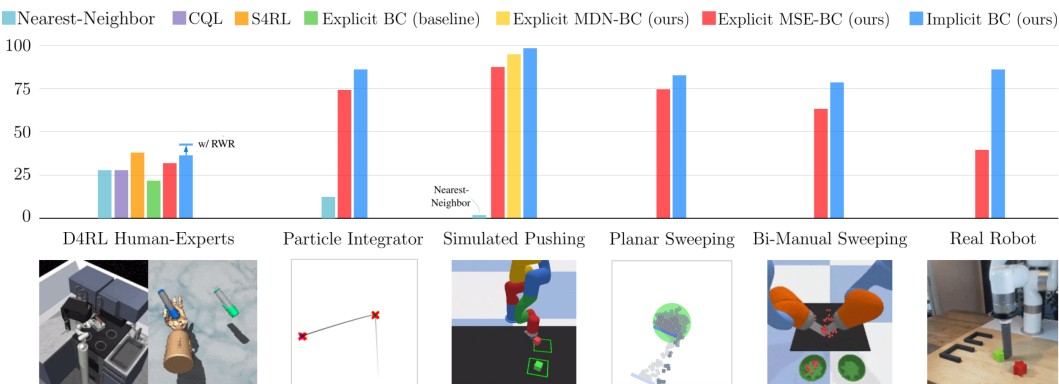

*Figure 5.* Comparisons between implicit and explicit policies across 6 various simulated and real domains (Table 1), including author-reported baselines on the human-expert D4RL tasks. See Appendix for full experimental protocol. Standard deviations are shown in Tables 2, 3, 4, 5, 6.

We evaluate implicit models for learning BC policies across a variety of robotic task domains (Fig. 5). The goals of our experiments are three-fold: (i) to compare the performance of otherwise-identical policies when represented as either implicit or explicit models, (ii) to test how well our models (both implicit and explicit) compare with author-reported baselines on a standard set of tasks, and (iii) to demonstrate that implicit models can be used to learn effective policies from human

| Benchmark | image input | human demos | unknown cardinality | multimodal solutions |
|---|---|---|---|---|
| D4RL Human-Experts | ✗ | ✓ | ✗ | ✗ |
| Particle Integrator | ✗ | ✗ | ✗ | ✗ |
| Block Pushing | ✓ | ✗ | ✗ | ✓ |
| Planar Sweeping | ✓ | ✓ | ✓ | ✓ |
| Bi-Manual Sweeping | ✓ | ✗ | ✓ | ✓ |
| Real Robot | ✓ | ✓ | ✗ | ✓ |

*Table 1.* Each benchmark is characterized by a unique set of attributes.

demonstrations with visual observations on a real robot. The following results and discussions are organized by task domain – each evaluating a unique set of desired properties for policy learning (Table 1). All tasks are characterized by discontinuities and require generalization (e.g., extrapolation) to some degree.

**D4RL [17]** is a recent benchmark for offline reinforcement learning. We evaluate our implicit (EBM) and explicit (MSE) policies across the subset of tasks for which offline datasets of human demonstrations are provided, which is arguably is the hardest set of tasks. Surprisingly, we find that our implementations of

| Method | Baselines | | | | Ours | | | |
| --- | --- | --- | --- | --- | --- | --- | --- | --- |
| | Nearest-Neighbor | BC (from CQL [26]) | CQL [26] | S4RL [27] | *Explicit* BC (MSE) | *Implicit* BC (EBM) | *Explicit* BC (MSE) w/ RWR [28] | *Implicit* BC (EBM) w/ RWR [28] |
| Uses data | (**o**,**a**) | (**o**,**a**) | (**o**,**a**,$r$) | (**o**,**a**,$r$) | (**o**,**a**) | (**o**,**a**) | (**o**,**a**,$r$) | (**o**,**a**,$r$) |

| *Domain* | *Task Name* | | | | | | | | |
| --- | --- | --- | --- | --- | --- | --- | --- | --- | --- |
| Franka | kitchen-complete | 1.92 ±0.00 | 1.4 | 1.8 | 3.08 | 1.76 ±0.04 | **3.37** ±0.19 | 1.22 ±0.18 | **3.37** ±0.01 |
| | kitchen-partial | 1.70 ±0.00 | 1.4 | 1.9 | **2.99** | 1.69 ±0.02 | 1.45 ±0.35 | 1.86 ±0.26 | 2.18 ±0.05 |
| | kitchen-mixed | 1.46 ±0.00 | 1.9 | 2.0 | | **2.15** ±0.06 | 1.51 ±0.39 | 2.03 ±0.06 | **2.25** ±**0.14** |
| Adroit | pen-human | 1908.0 ±0.0 | 1121.9 | 1214.0 | 1419.6 | 2141 ±109 | **2586** ±65 | 2108 ±58.8 | **2446** ±**207** |
| | hammer-human | -85.2 ±0.0 | -82.4 | 300.2 | **496.2** | -38 ±25 | -133 ±26 | -35.1 ±45.1 | -9.3 ±45.5 |
| | door-human | 91.8 ±0.0 | -41.7 | 234.3 | **736.5** | 79 ±15 | 361 ±67 | 17.9 ±13.8 | 399 ±34 |
| | relocate-human | -3.8 ±0.0 | -5.6 | 2.0 | 2.1 | -3.5 ±1.1 | -0.1 ±2.4 | -3.7 ±0.3 | **3.6** ±**2.5** |

*Table 2.* Baseline comparisons on D4RL [17] tasks with human-expert data. Results shown are the average of 3 random seeds, 100 evaluations each, with ± std. dev. Baselines from [26] and [27] didn't report standard deviations. See Appendix for more on experimental protocol.

both implicit and explicit policies significantly outperform the BC baselines reported on the benchmark, and provide competitive results with state-of-the-art offline reinforcement learning results reported thus far, including CQL [26] and S4RL [27]. By adding perhaps the simplest way to use reward information, if we prioritize sampling to be only the top 50% of demonstrations sorted by their returns (similar to Reward-Weighted Regression (RWR) [28]), this intriguingly generally improves implicit policies, in some cases to new state-of-the-art performance, while less so for explicit models. This suggests that implicit BC policies value data quality higher than explicit BC policies do. A simple Nearest-Neighbor baseline (see Appendix) performs better than one might expect on these tasks, but on average not as well as implicit BC.

While many of the D4RL tasks have complex high-dimensional action spaces (up to 30-D), they do not emphasize the full spectrum of task attributes (Table 1) we are interested in. The following tasks isolate other attributes or introduce new ones, such as highly stochastic dynamics (i.e., single-point-of-contact block pushing), complex multi-object interactions (many small particles), and combinatorial complexity.

**N-D Particle Integrator** is a simple environment with linear dynamics but where a discontinuous oracle policy is used to generate training demonstrations: once within the vicinity of goal-conditioned location (Fig. 5, shown for $N = 2$), the policy must switch to the second goal. The benefit of studying this environment is two-fold: (i) it has none of the complicating attributes in Table 1 and so allows us to study discontinuities in isolation, and (ii) we can define this simple environment to be in $N$ dimensions. Varying $N$ from 1 to 32 dimensions, but holding the number of demonstrations constant, we find we are able to train 95% successful implicit policies up to 16 dimensions, whereas explicit (MSE) policies can only do 8 dimensions with the same success rate. The Nearest-Neighbor baseline, meanwhile, cannot generalize, and only performs well on the 1D task (see Appendix for more analysis).

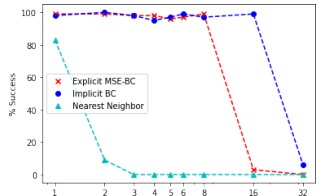

*Figure 6.* Comparison of policy performance on the $N$-D particle environment, 2,000 demonstrations each.

**Simulated Pushing** consists of a simulated 6DoF robot xArm6 in PyBullet [29] equipped with a small cylindrical end effector. The task is to push a block into the target goal zone, marked by a green square labeled on the tabletop. We investigate 2 variants: (a) pushing a single block to a single target zone, or (b) also pushing the block to a second goal zone (multistage). We evaluate implicit (EBM) and explicit (MSE and MDN [30, 31]) policies on both variants, trained

| Method | Single Target, states | Multi Target, states | Single Target, pixels |
| --- | --- | --- | --- |
| EBM | **100** ±0 | 99.0 ±0.0 | **100** ±0 |
| MDN | **100** ±0 | **99.7** ±**0.5** | 92.3 ±1.7 |
| MSE | 98.3 ±0.5 | 89.7 ±4.8 | 87.0 ±4.1 |
| Nearest-Neighbor | 4.0 ±0.0 | 0.0 ±0.0 | 4.3 ±1.9 |

*Table 3.* Results on simulated xArm6 pushing tasks, average of 3 random seeds, 100 evaluations each, with ± std. dev.

from a dataset of 2,000 demonstrations using a scripted policy that readjusts its pushing direction if the block slips from the end effector. Results in Table 3 show that all learning methods perform well on the single-target task, while MSE struggles with the slightly longer task horizon. For the image-based task, EBM outperforms both MDN and MSE. The failures of the Nearest-Neighbor baseline, with only 0-4% success rate, show that generalization is required for this task.

**Planar Sweeping** [32] is a 2D environment that consists of an agent (in the form of a blue stick) where the task is to push a pile of 50 - 100 randomly positioned particles into a green goal zone. The agent has 3 degrees of freedom (2 for position, 1 for orientation). We train implicit (EBM) and explicit (MSE) policies from 50 teleoperated human demonstrations, and test on episodes with unseen particle configurations. For the image-based inputs, we also test two types of encoders with different forms of dimensionality reduction: spatial soft(arg)max and average pooling over dense features (see Appendix for architecture descriptions).

For the state-based inputs, since the number of particles vary between episodes, we flatten the poses of the particles and 0-pad the vector to match the size of the vector at maximum particle cardinality.

The results in Table 4 (averaged over 3 training runs with different seeds) suggest that image-based EBMs outperform the best MSE architectures by 7%. Interestingly, image-based EBMs seem to synergize well with spatial soft(arg)max for dimensionality reduction, as opposed to pooling, which works best for MSE explicit policies. In both cases, state observations as inputs

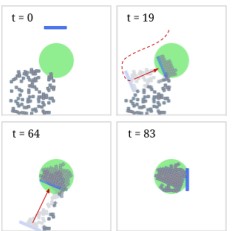

| Method | Input & Encoder | # ResNet layers | | |
|---|---|---|---|---|
| | | 8 | 14 | 20 |
| EBM | image + softmax | 78.7 ±4.9 | 82.1 ±0.9 | **82.6 ±3.1** |
| EBM | image + pool | 78.0 ±2.2 | 76.5 ±1.0 | 74.2 ±1.9 |
| EBM | state | 28.7 ±0.8 | 29.2 ±0.5 | 28.9 ±0.2 |
| MSE | image + softmax | 62.9 ±5.0 | 51.4 ±8.9 | 56.6 ±5.2 |
| MSE | image + pool | 75.6 ±1.3 | 73.9 ±1.7 | 74.8 ±1.2 |
| MSE | state | 28.9 ±0.2 | 28.2 ±0.4 | 27.8 ±0.3 |

*Figure 7 & Table 4.* Image-based implicit (EBM) policies outperform explicit (MSE) ones in learning to control the agent (blue) to sweep an unknown number of particles (gray) into a target goal zone (green). Trained on 50 human demonstrations.

do not perform well compared with image pixel inputs. This is likely because the particles have symmetries in image space, but not when observed as a vector of poses.

**Simulated Bi-Manual Sweeping** consists of two robot KUKA IIWA arms equipped with spatula-like end effectors. The task is to scoop up randomly configured particles from a $0.4m^2$ workspace and transport them into two bowls, which should be filled up equally. Successfully scooping particles and transporting them requires precise coordination between the two arms (e.g., such that the particles do not drop while being transported to the bowls). The action space is 12DoF (6DoF Cartesian per arm), and each episode consists of 700 steps recorded at 10Hz. Perspective RGB images from a simulated camera are used as visual input, along with current end effector poses as state input. The task is characterized by many mode changes and discontinuities (transitioning from scooping to lifting, from lifting to transporting, and deciding which bowl to transport to). EBM and MSE policies on the task use the best corresponding image encoder from the planar sweeping task. As shown in Table 5, our results show that EBM outperforms MSE by 14%.

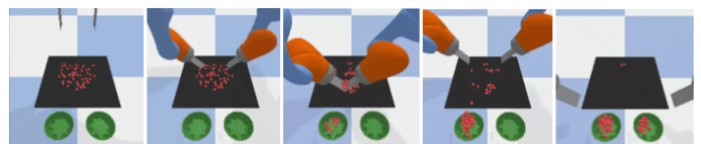

| Method | Input and Encoder | Success % |
|---|---|---|
| EBM | image + softmax | **78.2 ±2.7** |
| MSE | image + pool | 63.9 ±7.7 |

*Figure 8 & Table 5.* Image-based implicit (EBM) policies outperform explicit (MSE) ones in learning to control two robot arms (6DoF + 6DoF) with spatula-like end effectors to scoop up particles (red) from a workspace and equally distribute them across two bowls (green). Success % is the average ratio of particles successfully moved into the bowls across 10 rollouts over 3 different model seeds. Trained on 1,000 scripted demonstrations.

**Real Robot Manipulation**, using a cylindrical end-effector on an xArm6 robot (Fig. 9a), we evaluate implicit BC and explicit BC policies on 4 real-world manipulation pushing tasks: 1) pushing a red block and a green block into assigned target fixtures, 2) pushing the red and green blocks into either target fixture, in either order, 3) precise pushing and insertion of a blue block into a tight (1mm tolerance) target fixture, and 4) sortation of 4 blue blocks and 4 yellow into different targets. The observation input is only raw perspective RGB images at 5Hz, with task horizons up to 60 seconds, and teleoperated demonstrations.

| Task | Push-Red-then-Green | Push-Red/Green-Multimodal | Insert-Blue | Sort-Blue-from-Yellow |
|---|---|---|---|---|
| # demos | 95 | 410 | 223 | 502 |
| Avg. lengths ± std. [min, max] (seconds) | 19.1 ±2.5 [14.2, 25.1] | 19.0 ±3.1 [11.8, 28.1] | 22.1 ±5.5 [13.0, 43.5] | 45.2 ±8.2 [25.8, 60.5] |
| Success criterion | 1.0 if both blocks in target | 1.0 if both blocks in target | 0.5 for partial insert 1.0 for full insert | $\frac{1}{8}$ for each correct block in target |
| *Success avg. (%)* | | | | |
| Implicit BC (EBM) | **85.0 ±5.0** | **88.3 ±7.6** | **83.3 ±3.8** | **48.3 ±4.6** |
| Explicit BC (MSE) | 35.0 ±18.0 | 55.0 ±18.0 | 6.7 ±9.4 | 19.6 ±1.5 |

*Table 6.* Real-world robot results, success % shown is mean +/- std.dev (20 rollouts per seed, 3 seeds = 60 trials per method per task).

Across all four tasks, we observe significantly higher performance for the implicit policies compared to the explicit baseline. This is especially apparent on the pushing-and-oriented-insertion task (*Insert Blue*), which requires highly discontinuous behavior in order to subtly nudge enough, but not too far, the block into place (Fig. 9c). On this task we see the implicit BC policy has an *order of magnitude* higher success rate than the explicit BC policy. The sorting task in particular (*Sort-Blue-From-Yellow*, Fig. 9d) is our attempt to push the generalization abilities of our models, and we see a 2.4x higher success rate for the

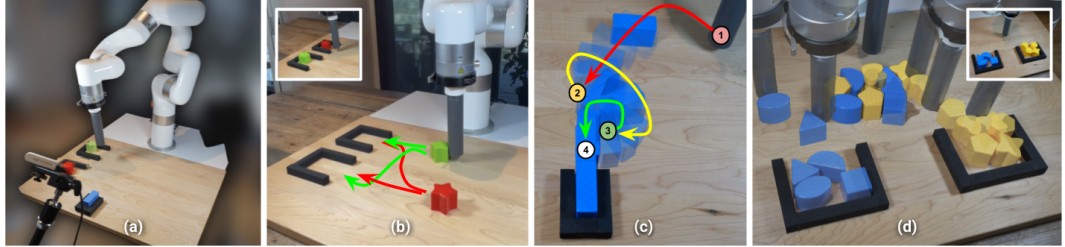

*Figure 9.* Results using our hardware configuration (a, see Appendix for full description) on real-world visual manipulation tasks, including (b) multi-modal targeted block pushing, (c) precise oriented insertion requiring 1mm precision, and (d) a combinatorially complex sorting task.

implicit policy. Note these experimental results are averaged over 3 different models, for each task, for each policy type. The red/green pushing tasks, including multi-modal variant (Fig. 9b), also show notably higher success rates for the implicit policies. These real-world results are best appreciated in our video.

## 5 Theoretical Insight: Universal Approximation with Implicit Models

In previous sections, we have empirically demonstrated the ability of implicit models to handle discontinuities (Section 3), and we hypothesized this is one of the reasons for the strong performance of implicit BC policies (Section 4). Two theoretical questions we now ask are: (i) is there a provable notion for *what class of functions* can be represented by implicit models given some analytical $E(\cdot)$, and (ii) given that energy functions learned from data may always be expected to have non-zero error of approximating any function, are there inference risks with large behaviour shifts resulting from a combination of $\mathrm{argmin}$ and spurious peaks in $E(\cdot)$? Recent work [33] has shown that a large class of functions (namely, functions defined by finitely many polynomial inequalities) can be approximated implicitly by $\mathrm{argmin}_{\mathbf{y}} g(\mathbf{x},\mathbf{y})$ using SOS polynomials to represent $g(\cdot)$. Here we show that for implicit models with $g_\theta$ represented by any continuous function approximator (such as a deep ReLU-MLP network), $\mathrm{argmin}_{\mathbf{y}} \, g_\theta(\mathbf{x},\mathbf{y})$ can represent a larger set of functions including multi-valued functions and discontinuous functions (Thm. 1), to arbitrary accuracy (Thm. 2). These results are stated formally in the following; proofs are in the Appendix.

**Theorem 1.** *For any set-valued function $F(\mathbf{x})\colon \mathbf{x}\in\mathbb{R}^m \to P(\mathbb{R}^n)\setminus\{\emptyset\}$ where the graph of $F$ is closed, there exists a continuous function $g(\mathbf{x},\mathbf{y})\colon \mathbb{R}^{m+n}\to\mathbb{R}^1$, such that $\underset{\mathbf{y}}{\mathrm{argmin}}\, g(\mathbf{x},\mathbf{y})=F(\mathbf{x})$ for all $\mathbf{x}$.*

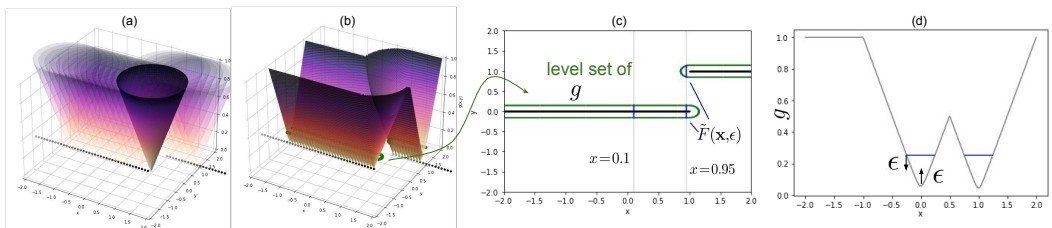

*Figure 10.* Visual explanation of the results presented in Thms. 1 and Thms. 2, the construction of a continuous function $g(x,y)$ for which $\mathrm{argmin}_y \, g(x,y)$ yields $f(x) = \{\{1,0\}$ if $x = 1, 1$ if $x > 1, 0$ otherwise$\}$. The function $g(\cdot)$ (b) is the minimum distance to the graph of $f()$, for example the infimum over a set of cones (a). The approximation guarantee (Thm. 2) can be visualized via the level-sets of $g(\cdot)$ (b,c), and a slice (d) of $g(\cdot)$. For more explanation, see the Appendix.

**Theorem 2.** *For any set-valued function $F(\mathbf{x})\colon \mathbf{x}\in\mathbb{R}^m \to P(\mathbb{R}^n)\setminus\{\emptyset\}$, there exists a function $g(\cdot)$ that can be approximated by some continuous function approximator $g_\theta(\cdot)$ with arbitrarily small bounded error $\epsilon$, such that $\hat{\mathbf{y}}=\underset{\mathbf{y}}{\mathrm{argmin}}\, g_\theta(\mathbf{x},\mathbf{y})$ provides the guarantee that the distance from $(\mathbf{x},\hat{\mathbf{y}})$ to the graph of $F$ is less than $\epsilon$.*

Of practical note, explicit functions ($F(\mathbf{x})$ in Thms. 1 and 2) with arbitrarily small or large Lipschitz constants can be approximated by an implicit function with bounded Lipschitz constant (see Appendix for more discussion). This means that implicit functions can approximate steep or discontinuous explicit functions without large gradients in the function approximator that may cause generalization issues. This is not the case for explicit continuous function approximators, which must match the large gradient of the approximated function. In both their multi-valued nature and discontinuity-handling, the approximation capabilities of implicit models are distinctly superior to explicit models. See Fig. 10 for visual intuition, and more discussion in the Appendix.

# 6 Related Work

**Energy-Based Models, Implicit Learning.** Reviews of energy-based models can be found in LeCun et al. [10] and Song & Kingma [19]. Du & Mordatch [12] proposed Langevin MCMC [11] sampling for training and implicit inference, and argued for several strengths of implicit generation, including compositionality and empirical results such as out-of-distribution generalization and long-horizon sequential prediction. A general framework for energy-based learning of behaviors is also presented in [34]. In applications, energy based models have recently shown state-of-the-art results across a number of domains, including various computer vision tasks [35, 36], as well as generative modeling tasks such as image and text generation [12, 37, 38]. Many other works have investigated using the notion of implicit functions in learning, including works that investigate implicit layers [39, 40, 41, 42, 43]. There is also a surge of interest in geometry representation learning in implicit representations [44, 45, 46, 47]. In robotics, implicit models have been developed for modeling discontinuous contact dynamics [48].

**Energy-Based Models in Policy Learning**. In reinforcement learning, [13] uses an EBM formulation as the policy representation. Other recent work [14] uses EBMs in a model-based planning framework, or uses EBMs in imitation learning [49] but with an on-policy algorithm. A trend as well in recent RL works has been to utilize an EBM as part of an overall algorithm, i.e. [15, 16].) Additionally, critic-only Q-learning [50] similarly defines the action mapping implicitly, but using reinforcement rather than supervised learning.

**Policy Learning via Imitation Learning.** In addition to behavioral cloning (BC) [1], the machine learning and robotics communities have explored many additional approaches in imitation learning [51, 52, 53], often in ways that need additional information. One route is by collecting on-policy data of the learned policy, and potentially either labeling with rewards to perform on-policy reinforcement learning (RL) [54, 55, 56] or labeling actions by an expert [2]. Distribution-matching algorithms like GAIL [7] require no labeling, but may require millions of on-policy environment interactions. While algorithms like ValueDice [57] implement distribution matching in a sample-efficient off-policy setting, they have not been proven on image-observations or high degree-of-freedom action spaces. Another route to using more information beyond BC is for the off-policy data to be labeled with rewards, which is the focus of the offline RL community [17]. All of these directions are good ideas. A perhaps not fully appreciated finding, however, is that in some cases even the simplest forms of BC can yield surprisingly good results. On offline RL benchmarks, prior works' implementations of BC already show reasonably competitive results with offline RL algorithms [17, 58]. In real-world robotics research, BC has been widely used in policy learning [4, 30, 5, 25]. Perhaps the success of BC comes from its *simplicity*: it has the lowest data collection requirements (no reward labels or on-policy data required), can be data-efficient [5, 25], and it is arguably the simplest to implement and easiest to tune (with fewer hyperparameters than RL-based methods).

**Approximation of Discontinuous Functions.** The foundational results of Cybenko [59] and others in Universal Approximation of neural networks have had foundational impact in guiding machine learning research and applications. Various approaches have been developed to approximate discontinuous functions [60, 61, 62, 63], which typically do not use neural networks. Also motivated by applications to modeling phenomena for robots, [64] develops theory of approximating discontinuous functions with neural networks, but the method requires a-priori knowledge of the discontinuity's location. Our work builds on the well-known and well-applied results in continuous neural networks, but through composition with $\arg\min$ provides a notion of universal approximation even for discontinuous, set-valued functions.

# 7 Conclusion

In this paper we showed that reformulating supervised imitation learning as a conditional energy-based modeling problem, with inference-time implicit regression, often greatly outperforms traditional explicit policy baselines. This includes on tasks with *high-dimensional action spaces* (up to 30-dimensional in the D4RL human-expert tasks), *visual observations*, and *in the real world*. In terms of limitations, a primary comparison with explicit models is that they typically require more compute, both in training and inference (see Appendix for comparisons). However, we have both shown that we can run implicit policies for real-time vision-based control in the real world, and algorithm complexity is simple compared to offline RL algorithms. To further motivate the use of implicit models, we presented an intuitive analysis of energy-based model characteristics, highlighting a number of potential benefits that, to the best of our knowledge, are not discussed in the literature, including their ability to accurately model discontinuities. Lastly, to ground our results theoretically we developed a notion of universal approximation for implicit models which is distinct from that of explicit models.

**Acknowledgments**

The authors would like to thank Vikas Sindhwani for project direction advice; Steve Xu, Robert Baruch, Arnab Bose for robot software infrastructure; Jake Varley, Alexa Greenberg for ML infrastructure; and Kamyar Ghasemipour, Jon Barron, Eric Jang, Stephen Tu, Sumeet Singh, Jean-Jacques Slotine, Anirudha Majumdar, Vincent Vanhoucke for helpful feedback and discussions.

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
