# OpenReview forum: "Implicit Behavioral Cloning"
_robot-learning.org/CoRL/2021/Conference — CoRL2021 Poster_

### Official Review · Reviewer_o4TR · 2021-07-22

**Originality:** Excellent
**Technical Quality:** Very Good
**Clarity Of Presentation:** Very Good
**Impact:** 4

**Recommendation:**

Strong Accept: I recommend accepting the paper and will argue for my recommendation even if other reviewers hold a different opinion.

**Summary:**

This paper proposes implict behavioral cloning, a modification to behavioral cloning that learns an implicit policy model (a scalar-valued function that accepts both state and action as inputs, and that must be minimized over actions to select an action) as opposed to the usual explicit policy model (a function that directly returns an action by taking a state as an input). The effectiveness of implicit policy models over explicit policy models is demonstrated over several toy function approximation problems, simulated domains, and physical real robot tasks. The paper also presents theoretical results that prove the effectiveness of implicit models in learning discontinuous, set-valued functions.

**Issues:**

Address the comments and questions in the weaknesses section above.

**Reviewer Expertise:**

Very good: Comprehensive knowledge of the area

**Strengths And Weaknesses:**

Strengths

The paper is very well-written and easy to follow. The empirical results are interesting and convincing. The toy function approximation results showcase the ability of implicit models to deal with discontinuous functions, multi-valued functions, and extrapolate favorably, when compared to implicit models. The results in simulated domains and real world tasks show that there can be substantial benfits to learning implicit models, especially when there are discontinuous or multimodal behaviors in the dataset.

The video results are extremely engaging (although narration would make it even better). The visualizations provided for the toy function regression examples and the multimodality and robustness qualitative real robot results were impressive.

Weaknesses

It would be nice to include a more thorough discussion of the limitations of implicit models compared to explicit models, especially in the context of real world visuomotor learning. There is some discussion in the appendix, but including some discussion in the main text could be useful to provide a more complete picture. For example, can implciit models only be run at 5 hz? What kind of slow down is there in policy execution times? Does it depend on the method used for optimizing the implicit function at evaluation time and the exact parametrization chosen for the implicit function (i.e. is there hope for reducing the slowdown)?

It could also be interesting to further evaluate the generalization capabilities of implicit models compared to explicit models directly on simulated or physical robot tasks, similar to the experiment presented in Figure 4, where implicit models can exhibit better generalization in lower data regimes. For example, can the implicit model exhibit better spatial generalization to learn a lifting task with less data? (see https://arxiv.org/abs/2103.00375 for an example lifting task). Even showing that implicit models can use less data than explicit models and achieve the same quality of performance as explicit models that use the full dataset on the proposed tasks would be interesting.

A few clarifying questions follow:
- How were policy checkpoints chosen for evaluation when training the explicit and implicit BC models? At least for explicit models, there can be significant variation in performance from epoch to epoch because a surrogate objective is being optimized instead of the true objective of interest (task performance).
- Were hyperparameters such as learning rate kept the same for implicit vs. explicit, or tuned for each method? Since the optimization procedure is different for implicit models, optimization hyperparameters should probably be tuned independently.
- How many demonstrations were collected and used for each real robot task?

**Summary Of Recommendation:**

This paper proposes an alternative policy parametrization for behavioral cloning that is well-motivated by toy examples and demonstrates impressive empirical performance in both simulated and real domains.

---

> ### Author Response · Authors · 2021-08-27
> **Authors' Response for Reviewer o4TR - Part 1**
>
> Thank you for your time in reviewing our paper, and thank you for your comments.  Here are some responses:
>
> Comment: **“The video results are extremely engaging (although narration would make it even better).”**
> - Response: Thank you.  We can make a narrated final version for a camera ready version, that is a good suggestion.
>
>
> Comment: **“The visualizations provided for the toy function regression examples and the multimodality and robustness qualitative real robot results were impressive.”**
> - Response: Thank you!
>
> Comment: **“It would be nice to include a more thorough discussion of the limitations of implicit models compared to explicit models, especially in the context of real world visuomotor learning. There is some discussion in the appendix, but including some discussion in the main text could be useful to provide a more complete picture.”**
> - Response:  This is a good suggestion.  In the main text, we now mention the primary limitations (increased compute for training and inference) in the Conclusion section, and we added a reference for the reader there to see the Appendix for more.  We expanded the discussion in the Appendix a bit more as well.
>
> Comment: (...talking about implicit vs. explicit) **“...can implicit models only be run at 5 hz? What kind of slow down is there in policy execution times? Does it depend on the method used for optimizing the implicit function at evaluation time and the exact parametrization chosen for the implicit function (i.e. is there hope for reducing the slowdown)?”**
> - Response: There wasn’t a 5 Hz limit, it could have been run faster – the reason is just that we find 5 Hz to be sufficient, and still reasonably fast, for the shown tasks.  You are right that inference time very much depends on the parameters used for optimizing the implicit function.  One interesting aspect of implicit models is that with iterative optimization methods, that they can do “anytime inference” – i.e., the iterative optimization can be cut early and at each iteration you can ask it for an answer and receive its best current guess.  Other parameters, such as the # of parallel samples for inference, can also be scaled down. So these provide, as you asked, a way to reduce the compute time, although less optimization provides less precise answers.  We’ve added some comparison numbers both on the training time and inference time into the appendix (Section C.2).
>
> Question: **“How were policy checkpoints chosen for evaluation when training the explicit and implicit BC models?”**
> - Answer: For all models (both explicit and implicit) we chose the # iterations at which the average evaluation performance of the 3 seeds was highest.  Appendix D.1 shows an example of evaluation over many returns.
>
> Question: **“Were hyperparameters such as learning rate kept the same for implicit vs. explicit, or tuned for each method? Since the optimization procedure is different for implicit models, optimization hyperparameters should probably be tuned independently.“**
> - Answer: Agreed, and accordingly they were tuned for the different methods. All swept and chosen parameters are noted in Appendix D.
>
> Question: **“How many demonstrations were collected and used for each real robot task?”**
> - Answer: This is noted at the top of Table 6.  Note that even our simplest real-world task is actually a sequence of two different block pushes, for which we used 95 demos.  We’ve also done tasks with fewer demonstrations, as few as 45 for a single “push block to location”.  We tried to really push our models with the challenging “Sort-Blue-from-Yellow” task, and so collected 502 demos for that task.

---

> > ### Author Response · Authors · 2021-08-27
> > **Authors' Response for Reviewer o4TR - Part 2**
> >
> > Comment: **“It could also be interesting to further evaluate the generalization capabilities of implicit models compared to explicit models directly on simulated or physical robot tasks, similar to the experiment presented in Figure 4, where implicit models can exhibit better generalization in lower data regimes.”**
> > - Response: Thanks, we agree that Figure 4 is a particularly useful example, and shows better generalization in low data regimes.  Another experiment that shows a notion of data complexity is the N-dimensional particle experiments, for which the # of demonstrations is held constant at 2,000, but N varies.  2,000 is a lot of demonstrations for the “1D” environment with a 4-dimensional observation space, but is not a lot of demonstrations for the “16D” environment which has a 64-dimensional observation space.  As the results show, we are able to get implicit models to generalize well in the “16D” environments, whereas we have not been able to get explicit models to generalize past 8D.  We’ve also added a Table to the Appendix which shows the # of demonstrations used in all of the different tasks – the set of evaluations covers different ends of the spectrum for small amounts of data (for example, just 25 demos on the D4RL-Adroit tasks with up to 30-dimensional action spaces) or large amounts of data.
> >
> > Thank you again for your time in reviewing our paper.  We hope these responses make sense.  We also note that you did give us high ratings for all of {Originality, Technical Quality, Clarity of Presentation, Impact}: {Excellent, Very Good, Very Good, and 4/4}, which we appreciate.  We are also glad that you appreciate the real world results!

---

> > > ### Comment · Reviewer_o4TR · 2021-08-28
> > > **Response**
> > >
> > > Thank you for your thorough reply - I found the additional details and comparisons in the appendix to be useful and provide a more complete picture, as well as address most of my comments. I would however like to ask a couple more follow-up questions.
> > >
> > > - How were success rates reported on the real robot tasks - was it just the policy checkpoint at the end of training? In simulation, it makes sense that the highest success rate checkpoint was reported, because it is relatively easy to evaluate the success rate of every checkpoint, but that can be far less feasible on the real robot.
> > > - How were demonstrations collected on the real robot, and how time-consuming was it, considering that you had to collect up to 500+ demonstrations for some tasks?

---

> > > > ### Author Response · Authors · 2021-08-29
> > > > **Responses to follow-up questions**
> > > >
> > > > Hi there, we're glad you appreciated the additional details and comparisons in the appendix, and that they addressed most of your comments. Here are some answers to your follow-up questions:
> > > >
> > > > Question: **"How were success rates reported on the real robot tasks - was it just the policy checkpoint at the end of training? In simulation, it makes sense that the highest success rate checkpoint was reported, because it is relatively easy to evaluate the success rate of every checkpoint, but that can be far less feasible on the real robot."**
> > > > - Answer: Yes, it's a good point that, in simulation, we can easily run many closed-loop evaluations to guide model selection, and that this isn't as easy in the real world.  For the real world experiments, as you expected, it was just the checkpoints at the end of training.  Please also note that even for the real world experiments, we used 3 training seeds per method, and we reported the standard deviation across the 3 seeds.
> > > >
> > > > Question: **"How were demonstrations collected on the real robot, and how time-consuming was it, considering that you had to collect up to 500+ demonstrations for some tasks?."**
> > > > - Answer: The method for doing the demonstrations is through teleoperation using a mouse-based interface -- this is mentioned in Section C.3.1 in the appendix.  Your question helps us realize that "teleoperation" wasn't mentioned in the main paper in the real world results section -- we can add this mention in.  For the total time of the demonstrations, this can be calculated from Table 6 by taking the avg. lengths of the demos in seconds, multiplied by the # of demos.  So for Push-Red-then-Green, this is 19.1 secs/demo x 95 demos = about 0.5 hours, or 1.37 hours for the Insert-Blue task.  All demos were collected by a single person, one of the authors.

---

> > > > > ### Comment · Reviewer_o4TR · 2021-08-29
> > > > > **Thanks**
> > > > >
> > > > > Thanks for addressing my questions and comments - I am increasing my score.

---

### Official Review · Reviewer_BwVg · 2021-07-23

**Originality:** Good
**Technical Quality:** Good
**Clarity Of Presentation:** Very Good
**Impact:** 4

**Recommendation:**

Weak Accept: I recommend accepting the paper, but will not argue for my recommendation if the majority of other reviewers have a different opinion.

**Summary:**

The paper proposes to perform Behavioral Cloning (BC) using implicit Energy Based Models (EBMs), and demonstrate the utility of the approach when compared to explicit models using simple feed-forward evaluation for inference and prediction. The authors compare to reasonable baselines on a large set of experiments, ranging from simple simulated environments to real-robot tasks. Although EBMs have been used extensively for on-policy RL problems,  the proposed approach applies strictly to the off-policy setting for imitation learning. A detailed analysis of the EBM modeling power is provided, along with an intriguing theoretical discussion of properties for function approximation.

**Issues:**

Please address the issues listed in the 'Weaknesses / Questions' portion of this review.

**Reviewer Expertise:**

Very good: Comprehensive knowledge of the area

**Strengths And Weaknesses:**

Strengths:

- The paper is well written and generally easy to follow.
- The idea is interesting and the results seem very promising.
- The analysis is quite extensive and detailed, and did a good job at highlighting useful properties of EBMs and justifying their utility for Imitiation Learning problems.
- A comprehensive set of experiments were conducted, which is to be commended. The toy problems help elucidate the benefits of using implicit models, while the simulated benchmark and real-robot demonstrate utility over baselines.
- The video was well made and informative.

Weaknesses / Questions:

- Extrapolation/Interpolation:  This property of the implicit approach is demonstrated on the simple 2D regression problems (Figure 3), and presumably should help in out-of-distribution modeling and generalization. However, this is not demonstrated for the higher-dimensional simulated/real robot experiments, as only performance metrics (such as success rates) are reported.
- Generalization: Related to the above point, it is unclear how well this method generalizes to new instances of the same task, or whether it is only over-fitting to data.  Given the model’s capacity to fit individual datum and “exhibit nearest-neighbor-like behavior” (line 71), combined with a large number of demonstrations (O(10^3)) for the high-dimensional tasks, there is concern that apparent generalization is actually a case of overfitting to densely-sampled data in action/Cartesian space. Showing that this is not the case, and that the method actually generalizes for high-dimensional tasks, could include tests with of different initial ‘piles’ or clusters of particles in the simulated Bi-manual sweeping experiment, for example.
- Regularization and Overfitting:  If generalization requires additional regularization terms in the loss function (such as  increasing entropy of the distribution), how might this affect the results compared to the explicit models?
- Video: It seems like the 2D experiments were not run to convergence for the Implicit model in many cases. Should we expect the EBM to collapse to the training data completely?
- Sample and  computational complexity: Given that the performance of the EBM requires additional sampling-based optimization for inference, how does the number of samples (or number of iterations for Langevin) scale with dimension and performance? One might presume this would be a bottleneck for high-dimensional problems (ex. joint space control of a quadruped robot).
- Real-world Experiments: the visual feedback policy operates at 5 Hz, but implicit model inference is 33 Hz (30ms). What is causing the bottleneck here? I find this surprising, but I may be missing something in the implementation description.


**Summary Of Recommendation:**

I find the work to be interesting and novel, and the scale of the experimental work is impressive. However, I have doubts regarding how well the approach generalizes. This is particularly the case in the manipulation experiments which use a large number of expert demonstrations in a relatively low-dimensional setting. As part of this, I am concerned that the model may be over-fitting to the data in many instances. If this is the case, then a discussion on appropriate methods for regularization may be necessary.

If the authors can demonstrate that this is in fact not the case, through convincing experiment and analysis (simulated results are fine), then I will happily eat crow and change my decision to a Weak Accept, at least. I would also like to have the other points in the review addressed.

---

> ### Author Response · Authors · 2021-08-27
> **Authors' Response for Reviewer BwVg - Part 1**
>
> Thank you for your time in reviewing our paper, and thank you for your well-organized comments.  Here are some responses:
>
> Comment from Summary: **“I find the work to be interesting and novel, and the scale of the experimental work is impressive. However, I have doubts regarding how well the approach generalizes. This is particularly the case in the manipulation experiments which use a large number of expert demonstrations in a relatively low-dimensional setting. As part of this, I am concerned that the model may be over-fitting to the data in many instances…. If the authors can demonstrate that this is in fact not the case, through convincing experiment and analysis (simulated results are fine), then I will happily eat crow and change my decision to a Weak Accept."**
> - Response: From your summary, It seems like the main point you are worried about is whether the models generalize well.  Interestingly, we think this is one of the strong points of the implicit models – their ability to generalize.  It sounds like you are mostly interested in new analysis or experimentation rather than our existing experiments, so we’ve added these three items for you:
>     1. You were concerned about “large numbers of demonstrations in low-dimensional settings”.  We’ve added a Table in the Appendix (Section C.1) which together shows the observation/state/action dimensionalities, along with the # of demonstrations, for all the policy learning experiments.  This highlights that some of our evaluations are in the very low-data regime, for example kitchen-complete uses only 19 demos, and the D4RL-Adroit tasks use as few as 25 demos but with 30-dimensional actions.
>     2. You were also concerned that the models may be overfitting in some instances, and in your “Generalization” point, you noted that the “nearest-neighbor-like behavior” might mean that the models are just memorizing the training data.  To address this concern, we’ve added exactly a “Nearest-Neighbor” baseline: a baseline that does indeed memorize the training data, and brute-force computes the closest observation in the training set, then performs lookup on the corresponding action (detailed explanation in a new Appendix section). Its performance is as follows:
>         - On the D4RL tasks, Nearest-Neighbor interestingly does better than one might think, but overall Implicit BC can do significantly better, for example (IBC vs. NN), 2586 +/-65 vs. 1908 +/-0 on pen-human, and 3.37 +/-0.19  vs. 1.92 +/-0.00 on kitchen-complete.
>         - On the N-d particle tasks: Nearest-Neighbor cannot generalize well at all: it only works well on the 1D environment.  Meanwhile IBC can generalize well at 16 dimensions.
>         - On the simulated pushing tasks:  Nearest-Neighbor models do not perform well, succeeding in only 4% of attempts on this task.  These tasks are goal-conditioned (the target location changes), and the implicit models instead learn how to generalize what the target location is, achieving 98-100% on these tasks.
>
>     3.  Although we can’t easily compute “extrapolation/interpolation” in high dimensions (but see our comment below on Figure 4) as is noted for example in  https://arxiv.org/abs/1603.04422, we added a new piece of analysis (Appendix C.6) for the N-d particle experiment which is “on average during evaluations, how close is the closest data point in the training set?”  This shows that as the dimensionality (N) of the environment increases, the training data gets significantly sparser.  This correlates with the Nearest-Neighbor models only performing well (as noted above) on the 1D environment, while IBC can generalize well in up to the 16D environment.
>
> Comment: **“Extrapolation/Interpolation: This property of the implicit approach is demonstrated on the simple 2D regression problems (Figure 3), and presumably should help in out-of-distribution modeling and generalization. However, this is not demonstrated for the higher-dimensional simulated/real robot experiments, as only performance metrics (such as success rates) are reported.”**
> - Response: In addition to Figure 3, please see Figure 4 in the “Visual Generalization” subsection, which shows extrapolation/interpolation on the coordinate regression task. The observations for this experiment are 96x96x3 images, i.e. a 27,648-dimensional observation space.  Figure 4 directly measures the “ability to fit” / “supervised learning error” on this higher-dimensional problem, and it shows extrapolation/interpolation through visualization of convex hull of the underlying state space (not observation space) and shows in Figure 4(d) that the implicit models generalize better in the low data regime.
>
> Comment: **“...visual feedback policy operates at 5 Hz, but implicit model inference is 33 Hz (30ms).”**
> - Response: Short answer: wasn’t a limit, the models could have been run at a faster rate and we’ve added timing information into the Appendix.  For more, please see our response to Reviewer o4TR who also asked this.

---

> > ### Author Response · Authors · 2021-08-27
> > **Authors' Response for Reviewer BwVg - Part 2**
> >
> > Comment: **“Regularization and Overfitting: If generalization requires additional regularization terms in the loss function (such as increasing entropy of the distribution), how might this affect the results compared to the explicit models?”**
> > - Response: For implicit models, we typically find good generalization without explicit regularization.  We think this is interesting, and perhaps the stochastic training process (which isn’t present in the training of the explicit models) provides a form of regularization.  We had previously experimented with several different types of regularization: DropOut, batch norm, layer norm – none of these improved our implicit models over the “unregularized” models.
> >
> > Comment: **“Video: It seems like the 2D experiments were not run to convergence for the Implicit model in many cases.”**
> > - Response: Please see Figure 3 for the converged implicit models.  We can update the video as well.
> >
> > Comment: **“Sample and computational complexity: Given that the performance of the EBM requires additional sampling-based optimization for inference, how does the number of samples (or number of iterations for Langevin) scale with dimension and performance? One might presume this would be a bottleneck for high-dimensional problems (ex. joint space control of a quadruped robot).”**
> > - Response: The results have demonstrated using Langevin on scaling from 1-D to 16-D action spaces on the N-D particle tasks, or scaling across the D4RL tasks, for which the kitchen envs have 9-D action spaces, and the Adroit envs have between 24-D and 30-D action spaces.  We can use the same hyperparameters despite the different dimensionalities.  A quadruped would typically be a 12D action space (i.e. a Boston Dynamics Spot has 3 motors per each 4 legs), which is in the same range as several of our tasks.  In our experience increasing the # of samples for Langevin is more associated with desired precision rather than dimensionality.  Langevin sampling has even been used for image generation (for example, https://arxiv.org/abs/2012.01316 for a recent one) and with a comparable number of Langevin iterations (on the order of 100) generates 128x128x3 images, which would be akin to a 49,152-D action space if it was a policy.
> >
> > Comments: **<all the Strengths you listed>**
> > - Response: Thank you!
> >
> > Thank you again for your time in reviewing our paper, and for your well-organized questions!  We hope this helps clarify some of your questions and concerns.

---

> > > ### Comment · Reviewer_BwVg · 2021-09-03
> > > **Reply to Authors' response.**
> > >
> > > I appreciate that the authors have addressed all my questions, and have provided acceptable explanations and additional results. I will increase my review score.

---

### Official Review · Reviewer_iNFj · 2021-07-24

**Originality:** Very Good
**Technical Quality:** Very Good
**Clarity Of Presentation:** Very Good
**Impact:** 4

**Recommendation:**

Weak Accept: I recommend accepting the paper, but will not argue for my recommendation if the majority of other reviewers have a different opinion.

**Summary:**

The authors present a new behavior cloning technique that allows for an implicit modeling of the task to improve the adherence of learned policies to demonstrations, thus improving performance. The authors achieve this by making a change to the policy representation such that it minimizes an energy function on the state and possible actions in the action space, and picks the lowest-energy action. The resulting policies are better at fitting diverse demonstrations.

**Issues:**

Please address data representation consistency

**Reviewer Expertise:**

Very good: Comprehensive knowledge of the area

**Strengths And Weaknesses:**

### Strengths:

S1) The paper is well structured and gets straight to the point about the innovation being discussed, which I appreciate.

S2) While the proposed innovation seems like a relatively minor iteration over existing methods, the utility of the change in service of better demonstration modeling is clear.

S3) I really like how a number of tasks of varying complexity are explored, especially the simpler problems which do well to establish the behavior of the proposed algorithm without results being obfuscated by complex dynamics.

### Weaknesses:

W1) Data representation could be improved - the number of seeds used for the data in each of the tables is not consistent, nor is the providing of variance information, which is only provided in Table 6.

W2) While I can see that the proposed algorithm is better at behavior cloning and cloning multiple value functions simultaneously, it is not clear to me that discontinuity, which this algorithm makes strides towards eradicating, is strictly bad. It seems to me that continuity could be useful at times, if disconnected demonstrations need to be connected by a single cohesive policy - I'm not saying I necessarily disagree with the authors on the utility of discontinuity, but it would help to have it better justified - particularly in cases where same states might command different actions depending on which demonstration is being cloned, which could potentially result in anomalous behavior in sequence.


### General notes:

This paper gave me some interesting things to think of - like what if we just trained RL actor-critic critic networks and took actions based on the argmax of the critic - though instability of value learning could be a problem there.

On the subject of existing imitation learning algorithms, papers such as DeepMimic and AMP: Adversarial Motion Priors may also bear mentioning as relevant to current state of the art - though they aren't quite behavior cloning so I don't hold that against the authors.

I will admit that I did not have time to verify the theory presented in Section 5 so it is possible I may have missed something there.



**Summary Of Recommendation:**

As mentioned, the data representation could use work on consistency.

---

> ### Author Response · Authors · 2021-08-27
> **Authors' Response for Reviewer iNFj**
>
> Thank you for your time in reviewing our paper, and thank you for your comments.  Here are a few responses to some of the comments you made:
>
> Comment: **“W2) While I can see that the proposed algorithm is better at behavior cloning and cloning multiple value functions simultaneously, it is not clear to me that discontinuity, which this algorithm makes strides towards eradicating, is strictly bad. It seems to me that continuity could be useful at times, if disconnected demonstrations need to be connected by a single cohesive policy…”**
> - Response: We agree, we do not think that discontinuity is inherently good or bad.  We do believe, however, that discontinuities are common in problems that we care about for robot learning agents – both due to discontinuous dynamics (like physical contact in the real world) and due to discontinuous policies (like an agent that, once it reaches something, switches its goal to something else.)  We also agree that continuity is useful at times, and it’s specifically the mix of dynamics and policies that mix both continuous and discontinuous that we think are most interesting.  Accordingly, many (but not all) of our experiments look at problems that have both continuous relationships for dynamics and policies, but also have moments of discontinuity.  An example is: in the simulated pushing task, the goal target location is a continuous variable which smoothly affects the policy, but when the block starts to slip, there is a discontinuous decision to adjust and fix it.
>
> Comment: **”...papers such as DeepMimic and AMP: Adversarial Motion Priors may also bear mentioning as relevant to current state of the art.”**
> - Response: Agreed, these are good references to add, we’ve added them in the updated draft.
>
> Comment: **”W1) Data representation could be improved - the number of seeds used for the data in each of the tables is not consistent, nor is the providing of variance information, which is only provided in Table 6.”**
> - Response: Thank you, in response to this we have added variance information in each table where appropriate.  Also we’ve updated so that all experiments which used multiple seeds used 3 seeds.  In terms of overall consistency of the experiments, some of the individual experiments explore different aspects of our approach.  For example: Figure 4 shows generalization and sample complexity; Table 2 shows D4RL tasks for which we can compare with author-reported baselines; Figure 6 showcases a special environment in which we can vary the dimensionality; and Table 4 explores different visual encoders of different model depths.  Overall however we have tried to summarize the high-level takeaways into Figure 5, which shows the per-domain average of different algorithms.  This is all to say, we hope you appreciate that we have thought carefully about consistency and data representation.  Certainly though, your suggestion of the added variance numbers is helpful – this was for sure a good suggestion, and we’ve added variance numbers to Tables 2, 3, 4, 5.
>
> Comment: **“...I'm not saying I necessarily disagree with the authors on the utility of discontinuity… but it would help to have it better justified - particularly in cases where same states might command different actions depending on which demonstration is being cloned, which could potentially result in anomalous behavior in sequence.”**
> - Response: Perhaps you are interested in instances which both have discontinuities, and also have multimodality.  For visualization of behavior on a simple 1D->1D regression problem, this is present in Figure 3a.  For cloning of scripted experts which we specifically scripted to have both discontinuities and multi-modalities, this is present in the bi-manual sweeping experiments.  It is also our opinion that many tasks with human experts will have demonstrations that have both multi-modalities and discontinuities.  For example in our real-world sorting task, there is a large amount of multi-modality (which block should be addressed first?) and also discontinuity (once a block has reached its target, move on to the next one).
>
>
> Thank you again for your time in reviewing our paper.  We hope these responses make sense.  We also note that you did give us high ratings for all of {Originality, Technical Quality, Clarity of Presentation, Impact} – {Very Good, Very Good, Very Good, and 4/4}, which we appreciate.  We’re glad that the paper “gave [you] some interesting things to think of” too.

---

### Author Response · Authors · 2021-08-31
**Authors' response to meta-review (recommended by PC to repost since the meta-review disappeared) - Part 1**

Thank you, Area Chair.  As you said, there were several clarifications requested from the reviewers, such as in complexity/scalability, experimental details, and limitations.  Here is a summary of our primary responses to the reviewer questions.  Additional detail and other topics are covered in the individual responses to reviewer posts.

*Generalization / sample complexity /  low-or-high dimensionality*: The primary question from Reviewer BwVg was around whether or not the models generalize well.  Relatedly, the question of “how well does this generalize?” also depends on the amount of samples used, and at some point if there is a sufficient density of training samples provided, then mere memorization of the training data is sufficient.  Relatedly as well, the ability of models to generalize may be very different at different scales of dimensionality, so the question of dimensionality is also tied together with generalization.  Reviewer o4TR also commented that they’d be interested in additional analysis on generalization, and Reviewer BwVg specifically asked for simulation experimentation on this topic.  Accordingly, we have 3 **new** pieces of analysis/experimentation:

1. **New Analysis: A Table Summarizing All Dimensionalities and # Demonstrations For All Tasks**  We think this will be especially helpful, both to the reviewers and also to other readers of the paper.  This new table, all in one place, helps highlight that the extensive experimentation we’ve provided covers a wide sampling of different parts of task parameter spaces for dimensionality and # of demonstrations.  For the “dimensionalities” of the environments, this is importantly composed separately of (i) the observation space (up to 129,600-D), (ii) the state space (up to 372-D), and the (iii) action space (up to 30-D) – these are all highlighted in this table.  Further, the table helps highlight that some experiments are in the very “low data regime”, for example some of the D4RL Adroit tasks have only 25 demonstrations, or kitchen-complete uses only 19.  This table is complemented by Figure 4(d), which graphs sample complexity on the visual coordinate regression task, and Figure 6, which graphs scaling dimensionality on the N-D particle task.  The new table is huge, so it can only fit in the Appendix – see Section C.1.
2. **New Experimentation: Nearest-Neighbor Baseline.**  We are especially excited about adding this baseline, which after much consideration we think is the best way to answer the question of: “wait, are the models just memorizing?”.  In 3D computer vision research, simple Nearest-Neighbor baselines have been shown to be surprisingly effective (for example https://arxiv.org/pdf/1905.03678.pdf), and we hope it becomes more common in policy learning experimentation as well.  Specifically in our case the Nearest-Neighbor baseline is what you get when you memorize the training data, and compute through brute force the closest observation in the training set, then perform lookup on the corresponding action (detailed explanation in a new Appendix section). As can be summarized in its inclusion in Figure 5, and detailed in inclusion in several Tables, overall Implicit BC is able to perform significantly better than the Nearest-Neighbor baseline in several key cases.  For example, IBC solves the simulated pushing tasks with 99-100% success, while Nearest-Neighbor gets 0-4%.
3. **New Analysis: Analysis of Data Sparsity**  One analysis we wanted to add was to quantify how much is the model generalizing, even in higher dimensional tasks?  For this we think the N-D particle tasks, for which we can systematically vary dimensionality, is a great tool.  To complement running the Nearest-Neighbor baseline on these tasks, we added a graph in the Appendix (Section C.6) which shows how the distance to the closest training point grows significantly, which helps explain why Nearest-Neighbor only does well on the 1-D task.  Meanwhile, Implicit BC is able to generalize well even at 16-D.

---

> ### Author Response · Authors · 2021-08-31
> **Authors' response to meta-review (recommended by PC to repost since the meta-review disappeared) - Part 2**
>
> *Adding variance numbers*: We used seeded evaluations, both for explicit models and implicit models, for most of our experiments, and it was a good request from Reviewer iNFj to add the variances from these numbers.  We’ve updated several tables in the paper .pdf with these numbers.  Overall, this helps give a sense of scale that the numbers are meaningfully different for Implicit BC, in many cases outperforming explicit baselines and offline RL baselines.
>
> *Limitations of Implicit Models*: This was a good suggestion from reviewer o4TR to add some discussion on their limitations into the main paper – we had previously included this in the Appendix,.  We’ve included a mention of this in the Conclusion of the main paper.  The primary comparison is that implicit models are more compute-intensive, both in training and inference.  We’ve added tables in the Appendix (Section C.2) that provide more insight into these tradeoffs.  Overall, however, despite the increased computational complexity, from a roboticist’s perspective we believe it’s important that we’ve demonstrated the models’ inference times are still low enough to be used in real-time visual control.  Typically with explicit models we use “batch sizes” for training, but not for inference.  With implicit models, we both use “batch sizes” for training and also for inference: much of the optimization is parallelized (across all the action samples),   This fits well with parallelizable compute often available for inference, like GPUs.
>
>
> We’d also encourage the Area Chair and reviewers to consider reviewing our full Contributions Statement which is included at the start of the Appendix.  We believe we have strong contributions in several areas: a novel formulation, extensive experimentation in simulation, real-world validation on real robots, and also theory.

---

### Meta-Review · Area_Chair_ZXMo · 2021-08-03

**Recommendation:** Accept (Poster)
**Confidence:** 4

**Metareview:**

=== comments before the discussion ===

This paper presents the advantages of energy-based models in the context of imitation learning. The proposed method is evaluated using D4RL benchmark tasks and real-robot experiments.
Reviewers agree that the paper is well-structured and the findings reported in the paper are interesting. However, reviewers requested some clarifications and discussions to improve the paper. Please answer the questions and address the concerns raised by the reviewers such as sample complexity/scalability, experimental details, and limitations of the proposed method.

=== comments after the discussion ===

This paper provides interesting results, and authors addressed the concerns raised by reviewers during the discussion period. The area chair recommends the acceptance of the paper.

---

> ### Author Response · Authors · 2021-08-27
> **Authors' Response to the Meta Review - Part 1**
>
> Thank you, Area Chair.  As you said, there were several clarifications requested from the reviewers, such as in complexity/scalability, experimental details, and limitations.  Here is a summary of our primary responses to the reviewer questions.  Additional detail and other topics are covered in the individual responses to reviewer posts.
>
> *Generalization / sample complexity /  low-or-high dimensionality*: The primary question from Reviewer BwVg was around whether or not the models generalize well.  Relatedly, the question of “how well does this generalize?” also depends on the amount of samples used, and at some point if there is a sufficient density of training samples provided, then mere memorization of the training data is sufficient.  Relatedly as well, the ability of models to generalize may be very different at different scales of dimensionality, so the question of dimensionality is also tied together with generalization.  Reviewer o4TR also commented that they’d be interested in additional analysis on generalization, and Reviewer BwVg specifically asked for simulation experimentation on this topic.  Accordingly, we have 3 **new** pieces of analysis/experimentation:
>
> 1. **New Analysis: A Table Summarizing All Dimensionalities and # Demonstrations For All Tasks.**  We think this will be especially helpful, both to the reviewers and also to other readers of the paper.  This new table, all in one place, helps highlight that the extensive experimentation we’ve provided covers a wide sampling of different parts of task parameter spaces for dimensionality and # of demonstrations.  For the “dimensionalities” of the environments, this is importantly composed separately of (i) the observation space (up to 129,600-D), (ii) the state space (up to 372-D), and the (iii) action space (up to 30-D) – these are all highlighted in this table.  Further, the table helps highlight that some experiments are in the very “low data regime”, for example some of the D4RL Adroit tasks have only 25 demonstrations, or kitchen-complete uses only 19.  This table is complemented by Figure 4(d), which graphs sample complexity on the visual coordinate regression task, and Figure 6, which graphs scaling dimensionality on the N-D particle task.  The new table is huge, so it can only fit in the Appendix – see Section C.1.
> 2. **New Experimentation: Nearest-Neighbor Baseline.**  We are especially excited about adding this baseline, which after much consideration we think is the best way to answer the question of: “wait, are the models just memorizing?”.  In 3D computer vision research, simple Nearest-Neighbor baselines have been shown to be surprisingly effective (for example https://arxiv.org/pdf/1905.03678.pdf), and we hope it becomes more common in policy learning experimentation as well.  Specifically in our case the Nearest-Neighbor baseline is what you get when you memorize the training data, and compute through brute force the closest observation in the training set, then perform lookup on the corresponding action (detailed explanation in a new Appendix section). As can be summarized in its inclusion in Figure 5, and detailed in inclusion in several Tables, overall Implicit BC is able to perform significantly better than the Nearest-Neighbor baseline in several key cases.  For example, IBC solves the simulated pushing tasks with 99-100% success, while Nearest-Neighbor gets 0-4%.
> 3. **New Analysis: Analysis of Data Sparsity.**  One analysis we wanted to add was to quantify how much is the model generalizing, even in higher dimensional tasks?  For this we think the N-D particle tasks, for which we can systematically vary dimensionality, is a great tool.  To complement running the Nearest-Neighbor baseline on these tasks, we added a graph in the Appendix (Section C.6) which shows how the distance to the closest training point grows significantly, which helps explain why Nearest-Neighbor only does well on the 1-D task.  Meanwhile, Implicit BC is able to generalize well even at 16-D.

---

> > ### Author Response · Authors · 2021-08-27
> > **Authors' Response to the Meta Review - Part 2**
> >
> > *Adding variance numbers*: We used seeded evaluations, both for explicit models and implicit models, for most of our experiments, and it was a good request from Reviewer iNFj to add the variances from these numbers.  We’ve updated several tables in the paper .pdf with these numbers.  Overall, this helps give a sense of scale that the numbers are meaningfully different for Implicit BC, in many cases outperforming explicit baselines and offline RL baselines.
> >
> > *Limitations of Implicit Models*: This was a good suggestion from reviewer o4TR to add some discussion on their limitations into the main paper – we had previously included this in the Appendix.  We’ve included a mention of this in the Conclusion of the main paper.  The primary comparison is that implicit models are more compute-intensive, both in training and inference.  We’ve added tables in the Appendix (Section C.2) that provide more insight into these tradeoffs.  Overall, however, despite the increased computational complexity, from a roboticist’s perspective we believe it’s important that we’ve demonstrated the models’ inference times are still low enough to be used in real-time visual control.  Typically with explicit models we use “batch sizes” for training, but not for inference.  With implicit models, we both use “batch sizes” for training and also for inference: much of the optimization is parallelized (across all the action samples),   This fits well with parallelizable compute often available for inference, like GPUs.
> >
> >
> > We’d also encourage the Area Chair and reviewers to consider reviewing our full Contributions Statement which is included at the start of the Appendix.  We believe we have strong contributions in several areas: a novel formulation, extensive experimentation in simulation, real-world validation on real robots, and also theory.

---

### Decision · Program_Chairs · 2021-09-13

**Decision:**

Accept (Poster)

**Comment:**

=== comments before the discussion ===

This paper presents the advantages of energy-based models in the context of imitation learning. The proposed method is evaluated using D4RL benchmark tasks and real-robot experiments.
Reviewers agree that the paper is well-structured and the findings reported in the paper are interesting. However, reviewers requested some clarifications and discussions to improve the paper. Please answer the questions and address the concerns raised by the reviewers such as sample complexity/scalability, experimental details, and limitations of the proposed method.

=== comments after the discussion ===

This paper provides interesting results, and authors addressed the concerns raised by reviewers during the discussion period. The area chair recommends the acceptance of the paper.